# Diatomites from the Iberian Peninsula as Pozzolans

**DOI:** 10.3390/ma16103883

**Published:** 2023-05-22

**Authors:** Jorge L. Costafreda, Domingo A. Martín, Beatriz Astudillo, Leticia Presa, José Luis Parra, Miguel A. Sanjuán

**Affiliations:** 1Escuela Técnica Superior de Ingenieros de Minas y Energía, Universidad Politécnica de Madrid, C/Ríos Rosas, 21, 28003 Madrid, Spain; domingoalfonso.martin@upm.es (D.A.M.); leticia.presa.madrigal@upm.es (L.P.); joseluis.parra@upm.es (J.L.P.); 2Laboratorio Oficial para Ensayos de Materiales de Construcción (LOEMCO), C/Eric Kandell, 1, 28906 Getafe, Spain; bastudillo@loemco.com; 3Department of Science and Technology of Building Materials, Civil Engineering School, Technical University of Madrid, 28040 Madrid, Spain; masanjuan@ieca.es

**Keywords:** diatomite, mortars, calcination, pozzolanicity, mechanical strength

## Abstract

The object of this work is to study and characterize diatomites from the southeast of the Iberian Peninsula to establish their character and quality as natural pozzolans. This research carried out a morphological and chemical characterization study of the samples using SEM and XRF. Subsequently, the physical properties of the samples were determined, including thermic treatment, Blaine particle finesse, real density and apparent density, porosity, volume stability, and the initial and final setting times. Finally, a detailed study was conducted to establish the technical properties of the samples through chemical analysis of technological quality, chemical analysis of pozzolanicity, mechanical compressive strength tests at 7, 28, and 90 days, and a non-destructive ultrasonic pulse test. The results using SEM and XRF show that the samples are composed entirely of colonies of diatoms whose bodies are formed by silica between 83.8 and 89.99% and CaO between 5.2 and 5.8%. Likewise, this indicates a remarkable reactivity of the SiO_2_ present in both natural diatomite (~99.4%) and calcined diatomite (~99.2%), respectively. Sulfates and chlorides are absent, while the insoluble residue portion for natural diatomite is 1.54% and 1.92% for calcined diatomite, values comparatively lower than the standardized 3%. On the other hand, the results of the chemical analysis of pozzolanicity show that the samples studied behave efficiently as natural pozzolans, both in a natural and calcined state. The mechanical tests establish that the mechanical strength of the mixed Portland cement and natural diatomite specimens (52.5 MPa) with 10% PC substitution exceeds the reference specimen (51.9 MPa) after 28 days of curing. In the case of the specimens made with Portland cement and calcined diatomite (10%), the compressive strength values increase even more and exceed the reference specimen at both 28 days (54 MPa) and 90 days (64.5 MPa) of curing. The results obtained in this research confirm that the diatomites studied are pozzolanic, which is of vital importance because they could be used to improve cements, mortars, and concrete, which translates to a positive advantage in the care of the environment.

## 1. Introduction

Diatomites are natural materials that form extensive deposits all over the planet; their chemical composition, based mainly on high silica contents, makes them multifunctional products. Nowadays, it is common to use diatomites in areas related to cement, concrete, and high-performance mortars. Yilmaz [1,2] and Janotka et al. [3] consider diatomites as amorphous and porous materials, and on this basis they have focused their research on the hydraulic effect produced by the standardized mixture of natural and calcined diatomites with Portland cement in varying proportions, resulting in an improvement in pozzolanic reactivity, a decrease in setting time, a reduction in the alkali, and a consequent increase in the normal mechanical strength of the mortars. Along the same lines, Mota and Chagas [4] as well as Paiva et al. [5] found that reducing the size of diatomite particles to 10 μm significantly increased the surface area, porosity, hydraulic activity, and mechanical strength of the diatomite-portlandite interface. Maina and Mbarawa [6] managed to improve the reactivity of lime by mixing it with diatomite, considering some factors such as temperature, diatomite-to-lime ratio, and the relationship between solid-liquid interfaces. Karaman et al. [7] and Gülbandilar and Koçak [8] showed that concretes and blocks made of diatomites not only have a high mechanical resistance to bending and compression but also a remarkable tensile strength and an obvious insulating property. Talero et al. [9] and Dedelouidis et al. [10] have significantly reduced the reaction products of reactive alumina, tricalcium aluminate, and tetracalcium aluminoferrite (C_4_AF) with the addition of diatomite and other varieties of pozzolan to Portland cement mixtures, making them more resistant to the effects of seawater. Gerengi et al. [11,12] proved that by subjecting concretes made from cements mixed with zeolite and diatomite to a solution of H_2_SO_4_, the effect of corrosion visibly decreased over a long period of time equal to 160 days. Degirmenci and Yilmaz [13] replaced part of the Portland cement with diatomite in variable proportions from 5 to 15%, resulting in an increase in mechanical strength to compression, greater resistance to ice-thaw cycles, decreased expansion of mortars, and an evident increase in sulfate resistance. Xu et al. [14,15] made mixtures of masonry mortar residues with diatomite parts and fly ash, achieving low-density mortars with high bending and compressive strengths as well as verifiable improvements in thermal and pozzolanic properties. In their extensive scientific review, Becerra-Duitama and Rojas-Avellaneda [16] highlight, along with other materials, the positive influence of diatomites on the cement-pozzolan interface, mainly in mechanical properties, durability, water absorption, and permeability. Pavlíková et al. [17] have focused their research on the improvement of mortars with diatomites intended for the repair and restoration of historic buildings and monuments and found that the optimal ratio of Portland cement to diatomite is 10%. Ince et al. [18] have monitored the mechanical properties of cement mortars previously reinforced with cellulose fibers, which have been introduced into the mixtures as a substitute for fine aggregate, resulting in a significant increase in mechanical compressive strength and an evident physical and chemical stability of the specimens. Rocha et al. [19] have improved the properties of geopolymers through a standardized mixture of diatomite, metakaolin, zeolite, and red sludge, achieving very stable reaction products from a physical-chemical point of view and with a high mechanical resistance. Tounchuen et al. [20] developed a complex mixture based on household waste gypsum and waste glass from the automotive industry with the addition of diatomite; according to their conclusions, the presence of diatomite in the mixture favors an increase in mechanical resistance and provides a marked impermeability as a result. Zhiqiang et al. [21] developed an eco-efficient concrete based on diatomite and iron ore waste in the proportions of 1, 5, 10, and 15%. The results of their research showed that the presence of diatomite favors the gain of mechanical strength, mainly in the proportion of 10% diatomite and 25% iron tailings. Finally, Yang et al. [22] have developed a structure composed of gypsum and diatomite to monitor humidity and temperature behavior in buildings, with satisfactory results and significant energy efficiency.

The main objective of this work is to determine the pozzolanic properties of diatomites from the southeast region of the Iberian Peninsula (Figure 1a,b) and establish their character as pozzolans capable of partially replacing Portland cement in mortars and concretes. To meet this objective, the work has been structured into three main phases. The first phase consists of the morphological, mineralogical, and chemical characterization of the samples using SEM and XRF; the second phase contains a characterization of the physical properties of the samples; and the third phase consists of a detailed qualitative and technical characterization of the samples through a quality chemical analysis. Additionally, we conducted a chemical analysis of pozzolanicity at 8 and 15 days, mechanical tests of compressive strength at 7, 28, and 90 days, and an ultrasonic pulse test in the mortar specimens.

The results obtained in this research could be a good guide for the improvement of traditional cements, which would have a positive impact on the quality of eco-efficient materials and the conservation of the environment.

## 2. Materials and Methods

### 2.1. Materials

To carry out this research, a volumetric sample of 50 kg was taken directly from a diatomite outcrop, courtesy of the company CEKESA. The sampling point is located in the vicinity of the municipalities of Hellín and Elche de la Sierra, both belonging to the Province of Albacete, in the southeast of the Iberian Peninsula (Figure 1a,b). Macroscopically, the rock color varies between white and beige yellow; it is not very dense and is marked by structures and microstructures of fine to very fine stratification. Its low cohesion, relative friability, and basal-type fracture shape are to be noted.

Prior to the analyses and tests, the samples were subjected to a rigorous, standardized preparation and conditioning process. In the first stage, the samples were ground up by a planetary ball mill, brand name Retsch, model PM 100, for 10 s at 250 rpm. Then, homogenization of the samples was carried out with an approved splitter. Once the total mass of the samples was homogenized, precise proportions were selected for each analysis and test.

### 2.2. Methods

#### 2.2.1. Morphological and Chemical Characterization Tests Using SEM and XRF


*Scanning electron microscopy:*


Scanning electron microscopy (SEM) was performed to determine the main morphological characteristics of the diatomite samples in their natural state and later in the calcined state. To meet this objective, the Hitachi S-570 electron microscope (Hitachi, Ltd., Hitachi-Shi, Japan) was used, which belongs to the Centralised Laboratory of the Escuela Técnica Superior de Ingenieros de Minas y Energía (Universidad Politécnica de Madrid). The microscope is equipped with a Kevex-1728 analyzer as well as a BIORAD Polaron (Bio-Rad, Hercules, CA, USA), a power supply for evaporation, and a Polaron SEM coating system. The study resolution is 3.5 nm, with an amplification of 200–103. The programs used are Winshell and Printerface.


*X-ray fluorescence:*


This study was conducted to determine the chemical composition of the natural diatomite sample (NDT). For this study, a part of the sample was prepared and mixed with flux at a diatomite-to-flux ratio of 0.3:5.5. Next, the sample was melted together with lithium tetraborate using a Philips Perl’3 induction apparatus (Philips, Amsterdam, The Netherlands). Subsequently, the sample was analyzed by wavelength dispersion X-ray fluorescence in the PANalytical equipment (Malvern Panalytical, Malvern, UK), which is provided with a rhodium tube.

#### 2.2.2. Physical Characterization Tests

Thermic treatment (TT) was carried out to remove as much moisture as possible from the sample. This test was conducted in two main phases: the first consisted of heating the sample to 105 ± 2 °C for 8 h, and for this, a Binder stove, model 90100101 ED 240, was used. In the second phase, the sample was calcinated with a Thermo Scientific Heraeus muffle furnace (Thermo Fisher Scientific, Waltham, MA, USA), model M 110 Muffle Furnace, at a normalized temperature of 900 °C and for an equivalent time of 1 h.

A granulometric analysis was performed to match the size of the diatomite sample particles as much as possible with the particles of the Portland cement, thus ensuring that the hydraulic reaction processes between the Portland and diatomite cement interfaces, as well as the reaction products, develop normally. To carry out this test, the Standard UNE-EN 196-6:2019 was used [24].

To determine the real density (RD), the guidelines of the Standard UNE-EN-80103:2013 [25] were followed by using the Micromeritics air pycnometer Accupyc 1330. In the case of apparent density (AD) and porosity (P), the Standard UNE-EN 1097-3:1999 was used [26].

The volume stability (VS) was carried out following the indications of the Standard UNE-EN 196-3:2017 [27]. The purpose of this method was to determine the volumetric expansivity of the mass of cement/diatomite studied.

The initial and final setting times (SFST) were determined according to the procedures indicated in the Standard UNE-EN 196-3:2017 [27].

#### 2.2.3. Qualitative and Technological Characterization Tests


*Chemical Analysis of Technical Quality:*


Chemical analysis of technological quality (CATQ) was performed to determine and establish the quality of the diatomite, both in its natural (NDT) and calcined (CDT) states, and to predict its behavior as pozzolanic material for partial replacement of Portland cement in pozzolanic cements, concretes, and mortars. This analysis was strictly carried out in accordance with the indications of the Standard UNE-EN 196-2:2014 [28].


*Chemical analysis of pozzolanicity:*


The purpose of the chemical analysis of pozzolanicity (CAP) is to determine the pozzolanic capacity of the samples using the Standard UNE-EN 196-5:2011 [29]. This analysis is based on the comparison between the concentration of the calcium ion (expressed as calcium hydroxide) present in the aqueous solution together with the hydrated mixture of NDT/PC and CDT/PC and the amount of calcium ion capable of saturating the solution with the same alkalinity. The concentration of the calcium ion in the solution must be less than the saturation concentration to have a satisfactory titration of the cement. In the development of this method, the concentrations of the hydroxyl ions [OH^−^] and CaO were determined by Equations (1) and (2), respectively.
(1)$$\left[{OH}^{-}\right]=\frac{1000\times 0.1\times V_{3}\times f_{2}}{50}=2\times V_{3\times }f_{2}$$
where:

[OH^−^] is the concentration in hydroxyl ions (mmol/L).

V_3_ is the volume of the hydrochloric acid solution (0.1 mol/L).

f_2_ is the factor of the hydrochloric acid solution (0.1 mol/L).
(2)$$\left[CaO\right]=\frac{1000\times 0.025\times V_{4}\times f_{1}}{50}=2\times V_{4}\times f_{1}$$
where:

[CaO] is the concentration of calcium oxide (mmol/L).

V_4_ is the volume of EDTA solution used in the titration.

f_1_ is the factor of the EDTA solution.


*Mechanical compressive strength test at 7, 28, and 90 days:*


In this paper, the mechanical behavior was studied in 6 mortar specimens made by a mixed formulation of diatomite in its natural (NDT) as well as calcined (CDT) states, each partially replacing Portland cement (PC) in proportions of 10, 25, and 40% (Table 1). Additionally, a reference specimen composed only of Portland cement (RMS) was developed in order to compare and monitor the mechanical behavior of the remaining specimens throughout the duration of the test. A mechanical compressive strength test was performed at 7, 28, and 90 days of curing. The steps taken in the development of this test have strictly followed the guidelines of the Standard UNE-EN 196-1:2018 [30]. In the preparation of the mortar specimens, 225 g of distilled water (DW) were used, as well as 1350 g of normalized sand (NS) as a fine aggregate.


*Test to determine ultrasonic pulse velocity in mortar specimens*
*:*


This method was developed to determine the propagation time of ultrasound waves as well as the ultrasonic pulse velocity (UPV) on mortar specimens made with mixtures of NDT/PC and CDT/PC, as well as the reference mortar specimen (RMS). The steps followed comply with the guidelines in the Standard UNE-EN ISO 16810 [31]. To determine the propagation time of the ultrasonic waves, CONTROLS-UPV E48 equipment and an Ultra-sehall-GEL gel were used.

Equation (3) was used for the calculation of UPV:(3)$${UPV}^{*}=\frac{D}{T}$$
where:

D is the distance measured in km.

T is time measured in µs.

The values of the UPV are expressed in km/s.

* [31].

## 3. Results and Discussion

### 3.1. Scanning Electron Microscopy

Figure 2A–D shows a series of microphotographs obtained by scanning electron microscopy (SEM) during the analysis of the natural diatomite sample (NDT). As can be observed, the sample consists of a compact agglomerate of frustules of organic origin consisting of the remains of multiform diatom algae skeletons. From a morphological point of view, the frustules have mostly radial or centric symmetry, are trelisoid, pennate, and, to a lesser degree, gonoidal. They also have a porous and striated appearance.

Figure 3A–D shows the appearance of the sample after being subjected to a controlled calcination process at 900 °C. Note how the original structures of the sample have been radically altered by heating, and the increase in pores is now more evident.

### 3.2. X-ray Fluorescence

Table 2 shows the results of the calculation of the chemical composition of the natural diatomite (NDT) and the Portland cement (PC) used in this work. The high content of SiO_2_ (83.8%) and CaO (5.2%) is noted as the most abundant major compounds in the composition of the sample. On the other hand, the SiO_2_-to-Al_2_O_3_ ratio is extremely high, while compounds such as sulfates and chlorides are well below the thresholds indicated in the standard. Loss on ignition (LOI) indicates a strong tendency of the sample for absorption and desorption and evident behavior as an open system. According to the data shown in Table 2, the chemical composition of the sample corresponds to that of typical diatomites [1,2,17].

### 3.3. Physical Tests

Table 3 and Figure 4, respectively, graphically represent the results of the particle size analysis with the particle size distribution of the natural diatomite sample (NDT), as well as the volume in which each granulometric range is grouped.

Table 4 shows the results of calculating the real density (RD), apparent density (AD), and porosity (P) of the sample. Additionally, Table 5 shows the results of volume stability (VS) as well as start and final setting times (SFST).

### 3.4. Chemical Analysis of Technical Quality

Table 6 shows the results of the chemical quality analysis of the diatomite samples, both in their natural (NDT) and calcined (CDT) states. A specific analysis of the NDT sample highlights the high content of SiO_2_ (89.99%), followed by CaO (5.80%). According to these results, it seems that the original sample is of high purity [32,33]. According to the values for reactive SiO_2_ (89.44%), it is certain that almost all the silica present in the sample can react in the solution. In the same way, the reactive CaO has reacted almost in its entirety, as can be seen when compared with the original content of the total CaO. The contents of Al_2_O_3_, MgO, and Fe_2_O_3_ are low and confirm that it is a typical diatomite; however, the average content of these compounds calculated for 15 different deposits in the Russian Federation [34] shows considerably high values for Al_2_O_3_ (6.05%) and Fe_2_O_3_ (2.56%) in these deposits, with MgO values (0.84%) being slightly comparable with those calculated in this research (Table 2 and Table 6). The high humidity value (13.90%) is indicative of the high absorption capacity of the sample, comparable to the work of Aksakal et al. [35]; this fact, together with the value calculated for the LOI (7.42%), establishes that the diatomite studied has evident properties as an ion exchanger [36].

The chemical composition of the calcined diatomite (CDT) sample does not show great differences in relation to NDT, but the SiO_2_ content is visibly elevated. Some authors, such as Fragoulis et al. [37], have shown that during the thermal processes of diatomites, SiO_2_ can be maintained and even increased; this is a direct effect of the calcination temperature. In this sample, almost all SiO_2_ is reactive, and only 0.64% of the portion is converted into insoluble residue (IR); however, unlike diatomite in its natural state, CDT shows low values of reactive CaO, which could be interpreted as a direct consequence of the heating process during which a part of the CaO could be recombined, forming calcium silicate, which is more stable and unable to react. This same calcination process also seems to produce a slight increase in insoluble residues, which could be considered reaction products with an inert nature. The drastic drop in LOI values indicates that virtually all volatile matter has been ejected from the sample during the calcination process [38].

According to the results discussed, the two samples meet the qualitative and technological requirements to be considered for use as pozzolans [29]. The low percentages of IR and chlorides, as well as the total absence of SO_3_, could make these materials suitable substitutes for Portland cement during the manufacture of cements with additives, pozzolanic cements, mortars, and concrete.

### 3.5. Chemical Analysis of Pozzolanicity

Figure 5a,b graphically show the results of the chemical analysis of pozzolanicity (CAP) performed on samples of natural diatomite (NDT) and calcined diatomite (CDT), which contain different proportions in mixtures of NDT/PC: 10–25–40% and CDT/PC: 10–25–40%, respectively. According to these results, the samples studied are established as pozzolanic, regardless of their natural or calcined state; in this way, the main objective of this research is achieved.

At 8 days of testing (Figure 5a), all analyzed samples are at a visibly deep position below the isothermal curve of CaO solubility, and this factor itself definitively determines the pozzolanic character of the sample [29]. Although the general trend in this period is a spatial concentration of all the results, the truth is that there are small variations that are relevant and should be highlighted. First of all, the samples of diatomite in its natural state (NDT) are located in an orderly manner and descend on the left side of the graph. The most pozzolanic is composed of the ratio NDT/PC: 10%, and the least pozzolanic is NDT/PC: 40%, which leaves sample NDT/PC: 25% in an intermediate position. Immediately to the right are the samples in a calcined state (CDT/PC: 10–25–40%), which follow a pattern similar to the previous ones. It is evident that the pozzolanic properties are increased from the CDT/PC formulations: 40% to the CDT/PC: 10%. Although the positions of both groups of samples in the graph are very close to each other, it can be established that heat treatment significantly improves the pozzolanic character [39].

At 15 days (Figure 5b), the pozzolanic reactivity of all samples continues to manifest, even though they maintain some spatial proximity. A noteworthy fact, however, is the linear pattern the samples follow, whereby the most pozzolanic (CDT/PC: 10%) is in the deepest position on the right-hand side of the graph, while the least pozzolanic (CDT/PC: 40%) is located towards the extreme left. According to the results, it is evident that there are two key factors that determine the intensity with which the pozzolanic reactivity manifests: the partial replacement of Portland cement by diatomite and the degree of calcination applied to said sample.

Some authors, such as Tironi et al. [40], have emphasized this fact in their research with clay and kaolinitic materials of inorganic origin. However, other criteria discussed and verified in this work are accepted as valid, such as chemical composition, absorption capacity, morphological and surface characteristics, as well as porosity. Costafreda [38] has taken some of these factors into account in his previous research.

### 3.6. Mechanical Compressive Strength Tests at 7, 28, and 90 Days

Figure 6a,b graphically show the results obtained by the mechanical strength test (MST) at 7, 28, and 90 days. According to the way in which the samples are arranged in both graphs, it seems that the presence of diatomite in the formulations of the specimens, far from interfering negatively, rather causes a visible increase in mechanical resistance to compression from 7 to 90 days of curing. Based on the above, not only is the pozzolanic character established once again in the samples studied, but also their ability to replace Portland cement and provide mechanical resistance to the specimens.

A detailed analysis of Figure 6a shows some remarkable evidence: (1) the greater the replacement of PC by diatomite (NDT/PC: 40%), the mechanical strength values increase very slowly, which is typical in pozzolanic reactions [41]; (2) when substitution decreases to 25% (NDT/PC: 25%), it seems that pozzolanic reactivity is reinforced, a fact that becomes more obvious after 28 days of curing; and (3) when the substitution is 10% (NDT/PC: 10%), the mechanical strength provided by the diatomite is comparatively similar to the mechanical strength determined for the reference mortar specimen (RMS). Cases similar to those just discussed have been described by several researchers on other types of pozzolanic materials, among which the works of Küçükyıldırım and Uzal are cited [42].

According to Figure 6b, it is evident that the previous pattern is repeated, where significant increases in mechanical strength can be verified throughout the three test periods. In addition to the three facts discussed in the previous paragraph, in this case a fourth fact is revealed: the effect produced by the heat treatment on the sample. Indeed, this factor seems to cause a strengthening of the pozzolanic capacity of diatomites once calcined, which manifests in the form of significant increases in mechanical resistance. Note how these values increase mainly after 28 days, highlighting the fact that in this period, the specimens made with the CDT/PC: 10% ratio quantitatively exceed the mechanical strength value of the reference specimen (RMS). Some authors, such as Martín et al. [43], have demonstrated the veracity of these arguments in their research with other varieties of pozzolans.

Based on the above criteria and the data contained in Table 7, the Resistant Activity Index (RAI) has been calculated using Equation (4):(4)$${RAI}^{*}=\frac{V_{1\times f_{\left(\%\right)}}}{V_{2}}$$
where:

V_1_ is the mechanical compressive strength (MPa) of mixed mortar specimens produced according to NDT/PC and CDT/PC formulations.

V_2_ is the mechanical compressive strength (MPa) of reference mortar specimens.

f_(%)_ is the percentage factor with a value of 100%.

* RAI ≥ 75% [30].

**Table 7 materials-16-03883-t007:** Calculation of the Resistant Activity Index (RAI) values from the mechanical strengths of mortar specimens made with NDT and CDT in proportions of 10, 25, and 40%.

Sample	7 Days of Curing		28 Days of Curing		90 Days of Curing	
	Compressive Strength (MPa)	RAI * Calculated (%)	Compressive Strength (MPa)	RAI Calculated (%)	Compressive Strength (MPa)	RAI Calculated (%)
RMS	42.7	-	51.6	-	57.8	-
NDT-40	14.3	33.5	25.9	50.1	27.9	48.2
NDT-25	22.8	53.3	35.8	69.3	43.5	75.2
NDT-10	41.0	96.0	52.4	101.5	56.6	97.9
CDT-40	17.2	40.2	28.1	54.4	33.0	57.0
CDT-25	24.6	57.6	41.1	80.0	44.4	77.0
CDT-10	41.4	97.0	54.0	104.6	64.2	111.0

* Resistant Activity Index.

According to the data shown in Table 7, an increase in RAI is observed in the set of mortar specimens analyzed from 7 to 90 days; however, it seems that only two samples (NDT-10% and CDT-10%) are able to exceed 75% of the RAI after 7 days of curing, which does not mean a negative factor when taking into account that pozzolans are characterized by slowing down the setting process and increasing resistance in the initial stages of curing, between 7 and 28 days [38,43].

Analysis of the 28-day estimate reveals not only an unavoidable increase in resistant activity but also that the NDT-10% and CDT-10% samples exceed 100% of the RAI, while the CDT-25% sample now reaches a RAI equal to 80%. After 90 days of curing, in addition to the three samples indicated above, the NDT-25% sample has exceeded the lower limit of the RAI. The conclusions obtained through this analysis allow us to confirm once again the pozzolanic behavior of the samples analyzed.

### 3.7. Ultrasonic Pulse Velocity Determination

Table 8 shows the results obtained by the ultrasonic pulse velocity (UPV) determination test performed on mortar specimens made with the formulations NDT/PC: 10–25–40% and CDT/PC: 10–25–40%. The first point to highlight is the direct correspondence between the percentage of diatomite in the specimens and the propagation time of the ultrasonic wave, meaning that the larger the portion of diatomite, the longer the ultrasonic wave takes to cover the space between both ends of the specimen. Note the values obtained for the six specimens studied (NDT-40-25-10 and CDT-40-25-10). On the other hand, the data in Table 8 show an inversely proportional correspondence between the speed of the UPV and the propagation time of the wave, as the higher the time values, the lower the speed of the ultrasonic pulse, and vice versa. Finally, it is observed that only the NDT-10 and CDT-10 samples significantly approximate their values to those of the reference specimen (RMS).

These arguments establish that the factors depend on the chemical and physical properties of diatomites, mainly porosity, particle size, and total and reactive SiO_2_ content. Some authors, such as Presa et al. [44], not only mention this fact but have also made accurate predictions about the durability and mechanical resistance of mortar specimens by determining the UPV, showing the effectiveness of this non-destructive test.

## 4. Conclusions

Now that the results of this research have been stated and discussed, the following conclusions are established:The samples of diatomites studied, both in their natural (NDT) and calcined (CDT) states, behave similar to typical natural pozzolans; the features that potentially affect this fact are: their chemical composition, mainly the high contents of total SiO_2_ and reactive SiO_2_, porosity, absorption capacity, tendency to loss on ignition, and the active surface of diatomite particles.Pozzolanic reactivity visibly increases after calcination of the diatomite, which may be due to an increase in the degree of porosity of the sample and an increase in amorphous silica from total SiO_2_ after heating.The presence of NDT and CDT provides effective volume stability to mixed mortar paste and considerable durability to hardened specimens.The setting start and final times of the paste of mortars containing NDT and CDT tend to increase relative to the fresh mortar reference, especially when the portion of diatomite is between 25 and 40%; however, these values are drastically reduced when the proportion is 10%.The mechanical resistance to compression increases visibly between 7 and 90 days of curing, mainly in those specimens that contain NDT and CDT at 10 and 25%. In some cases, these increases exceed the value of the mechanical compressive strength of the reference specimen. This same trend has been determined with the calculation of the Resistant Activity Index.Specimens made with NDT and CDT slow down the rate of the ultrasonic pulse through the specimens, which can be a direct consequence of the high silica content and porosity of the sample, factors that produce a strong acoustic impedance. This characteristic could be a predictive factor when formulating lighter mortars and concretes with insulating properties, both thermal and acoustic.

## Figures and Tables

**Figure 1 materials-16-03883-f001:**
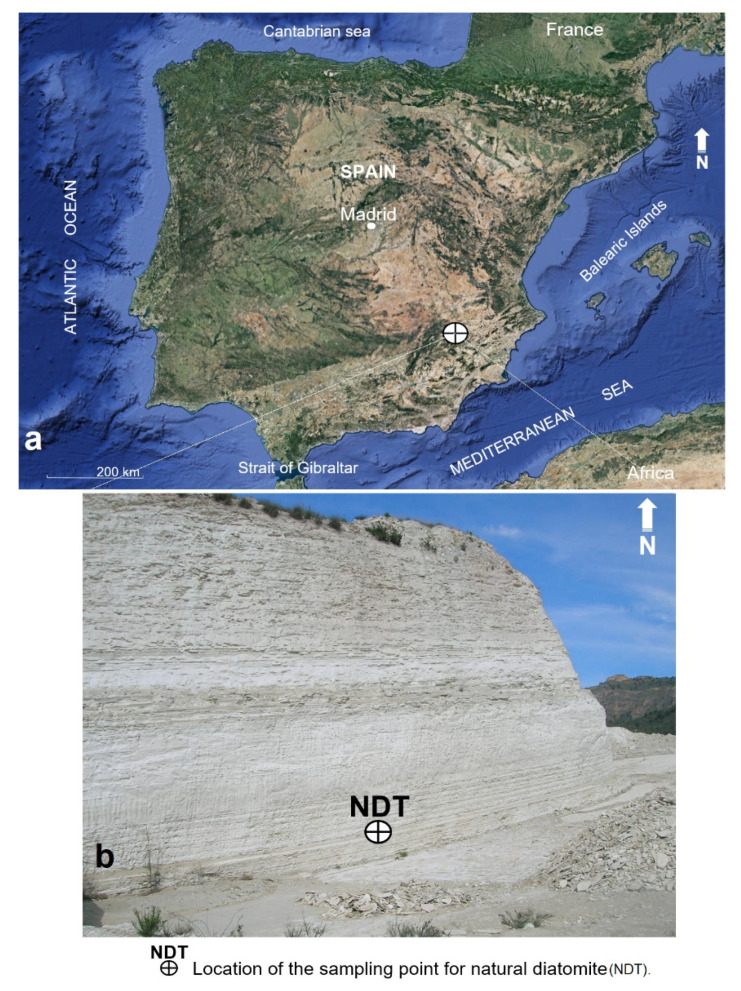
Research area location (**a**) [23] and detail of the point of extraction of the volumetric sample of natural diatomite (**b**).

**Figure 2 materials-16-03883-f002:**
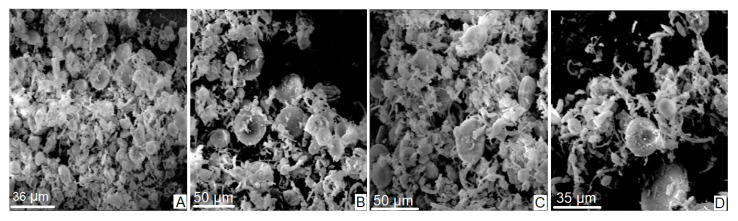
Microphotographs (**A**–**D**) taken from the volumetric sample of natural diatomite (NDT) by SEM.

**Figure 3 materials-16-03883-f003:**
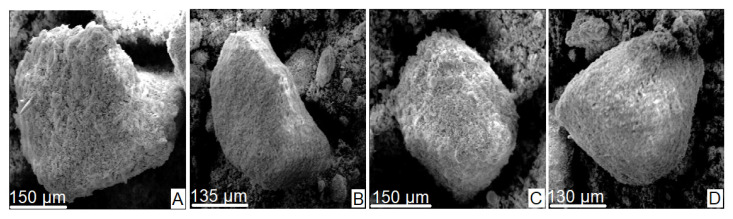
Microphotographs (**A**–**D**) taken from the volumetric calcined diatomite (CDT) sample by SEM.

**Figure 4 materials-16-03883-f004:**
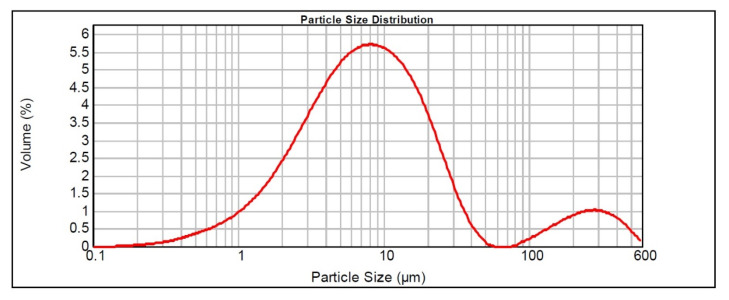
Particle size distribution obtained by granulometric analysis of the natural diatomite sample.

**Figure 5 materials-16-03883-f005:**
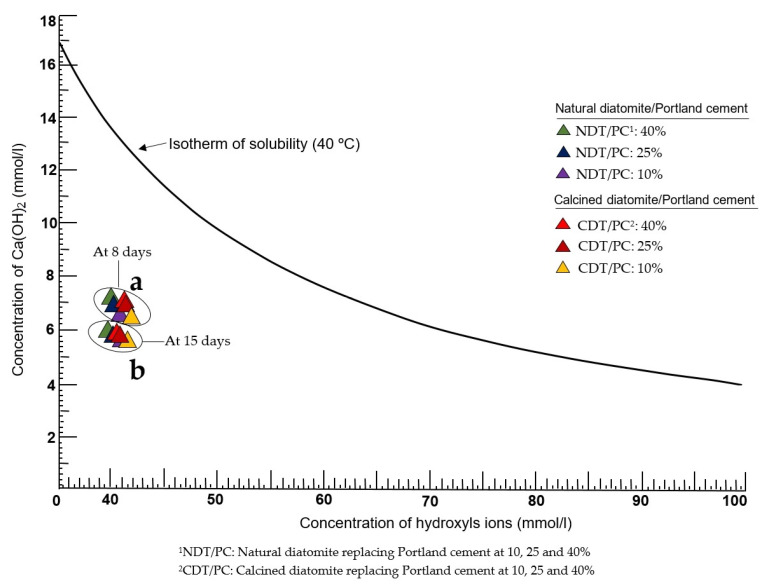
Pozzolanic behavior of natural and calcined diatomite samples at (**a**) 8 days and (**b**) 15 days.

**Figure 6 materials-16-03883-f006:**
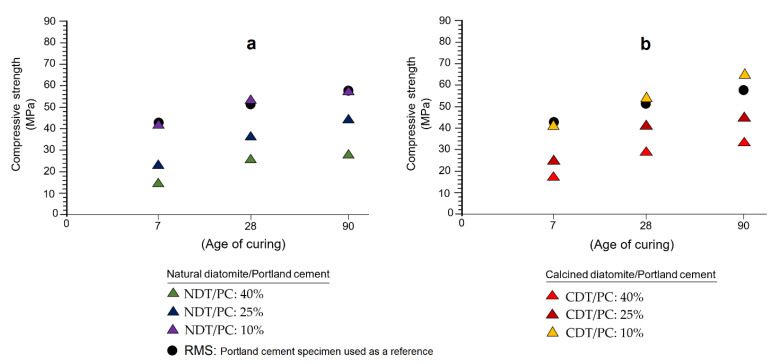
Results of the tests of mechanical strength to compression obtained from the study of specimens made with natural diatomite (**a**) and calcined (**b**) at 7, 28, and 90 days.

**Table 1 materials-16-03883-t001:** Dosage formulation of mixtures of natural diatomite (NDT) and calcined (CDT), partially replacing Portland cement (PC).

Sample	Proportion (Ratios)			Temperature of Calcination (°C)	(B.P.F.) ^6^ (cm^2^/g)
	NDT ^1^/CDT ^2^:PC ^3^ (%)	NS ^4^ (g)	DW ^5^ (g)		
NDT-40	40:60	1350	225	-	5416
NDT-25	25:75			-	
NDT-10	10:90			-	
CDT-40	40:60			900	
CDT-25	25:75			900	
CDT-10	10:90			900	

^1^ Natural diatomite; ^2^ Calcined diatomite; ^3^ Portland cement; ^4^ Normalized sand; ^5^ Distilled water; ^6^ Blaine particle fineness.

**Table 2 materials-16-03883-t002:** Results of chemical analyses obtained by X-ray fluorescence.

Sample	Compounds in % Weight											
	SiO_2_	Al_2_O_3_	Fe_2_O_3_	CaO	TiO_2_	SO_3_	K_2_O	MgO	P_2_O_5_	Na_2_O	Cl	LOI *
**NDT**	83.8	1.0	0.5	5.2	0.06	0.09	0.13	0.38	0.06	0.02	0.02	8.3
**PC**	17.47	5.57	3.39	64.01	0.33	4.0	1.39	0.64	0.07	0.09	-	2.41

* Loss on Ignition.

**Table 3 materials-16-03883-t003:** Details of the results of the granulometric analysis performed on the natural diatomite sample.

Size (µm)	Retained (%)	Passing through (%)	Percentage (%)	Size (µm)	(B.P.F.) ^1^ (cm^2^/g)
32	10.4	89.5801	10	1.876	5416
45	8.6	91.4355	50	7.807	
63	8.4	91.6265	63.2	11.17	
90	8.3	91.6711	90	33.525	

^1^ Blaine particle fineness. Type of distribution: volumen. Average diameter D [4,3]: 31.534 μm. Distribution width (10–90%)/50%: 4.054. Mode: 8.137 μm.

**Table 4 materials-16-03883-t004:** Results of the calculation of the real density, apparent density, and porosity of the natural diatomite sample (NDT).

Sample	RD ^1^ (g/cm^3^)	AD ^2^ (g/cm^3^)	Porosity (%)
NDT	1.98	0.173	0.912

^1^ Real density; ^2^ Apparent density.

**Table 5 materials-16-03883-t005:** Results of the volume stability test and the start and final setting times of natural diatomite (NDT) and calcined diatomite (CDT) samples.

Sample	Volume Stability (mm)			Start and Final Setting Time	
	A ^1^	C ^2^	C-A	Start (min)	Final (min)
RMS ^3^	0	0	0	170	230
NDT-40 ^4^	0.5	1	0.5	210	295
NDT-25	0	0	0	195	265
NDT-10	0	0	0	130	175
CDT-40 ^5^	0	0	0	205	265
CDT-25	0	0	0	200	245
CDT-10	0	1	1	155	220

^1^ Distance measured between the tips of the needles after resting the Le Chatelier equipment for 24 h ± 30 min at 20 ± 1 °C in the wet chamber; ^2^ Distance measured between the tips of the needles after allowing the Le Chatelier equipment to cool to room temperature; ^3^ Reference mortar sample; ^4^ Natural diatomites at 40, 25, and 10%; ^5^ Calcined diatomites at 40, 25, and 10%.

**Table 6 materials-16-03883-t006:** Results of the chemical composition and pozzolanic quality of the samples studied.

Compounds	NDT ^1^ (%)	CDT ^2^ (%)	Limit Allowed * (%)
Total SiO_2_	89.99	84.64	-
Reactive SiO_2_	89.44	84.00	>25
Total CaO	5.80	5.48	-
Reactive CaO	4.33	0.89	-
Al_2_O_3_	0.92	0.92	<16
MgO	0.38	0.39	<5
Fe_2_O_3_	0.42	0.28	-
SO_3_	-	-	<4
Humidity	13.90	-	-
IR ^3^	1.54	1.92	<3
LOI ^4^	7.42	1.96	-
Cl ^5^	0.01	0.01	<0.1
SiO_2_/(CaO + MgO)	19.11	66.13	>3.5

^1^ Natural diatomite; ^2^ Calcined diatomite; ^3^ Insoluble residue; ^4^ Loss on ignition; ^5^ Chlorides; * [28].

**Table 8 materials-16-03883-t008:** Determination of the ultrasonic wave propagation time and ultrasonic pulse velocity (UPV) of the samples.

Samples	Ultrasonic Wave Propagation Time (µs)	Value of UPV Calculated (km/s)
RMS	36.33	4.40
NDT-40	44.60	3.59
NDT-25	40.83	3.92
NDT-10	37.06	4.25
CDT-40	44.13	3.63
CDT-25	41.43	3.86
CDT-10	37.43	4.27

## Data Availability

Not applicable.

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
