# Peer review of "Diatomites from the Iberian Peninsula as Pozzolans"

_materials, 2023, doi:10.3390/ma16103883_

Round 1
Reviewer 1 Report
Diatomite is a rock and I found many articles describing this formation. A description of formation and the rock must be included. Lines 117-120, the description of the equipment is not clear. Why did not the authors determine the mineralogical composition of the rock ? It is important to know what mineral (SiO2) is present, and the same thing for the calcium carbonate. The behaviour of different minerals with the same chemical composition is not the same. In many some cases the information given in the tables is the same than in the figure. This is not necessary. Waves propagation time is not a useful data and, for constant length of the samples, it gives the same information than the velocity. Please remove. I do not understand the granulometric analysis. I supposed the rock has been grinded to produce the aggregates that have used. I think we cannot say that is the size of diatomite but the size of the aggregates. Table 4. Mass and volume are not necessary, they depend on the size of the samples. Only density and porosity are useful data. What about the mineral formed in the mortars ? We do not have any information about that. Line 321. There is not Figure 5.b. Line 343. The results show that mechanical resistance decreases with the amount od diatomite. May the authors explain the meaning of this sentence? Which is the porosity of different mortars ? I think the conclusion and discussion is not clear. Of course, mechanical strength increases with carbonation (time) but for all the times, except one sample with 10 calcinated diatomite, mechanical strength decreases when substituting Portland cement by diatomite. Conclusions must be revised.
Moderate editing of English language
Author Response
Dear Reviewer:
Thank you very much for your suggestions and comments, which have been very helpful and accurate.
Please find enclosed the answers to all your questions.
Kind regards;
Dr. Jorge L. Costafreda.

Reviewer 2 Report
The paper is clearly and well written.
I have several remarks:
Line 70. Diatomite/PC is 10/90?
Lines 179-180. According to formular the solution is 0.5V4f1
Lines 232-234 It is not clear. Figures 2 and 3 have to be presented in the same scale.
Table 3. Distribution width (10% - 90%)/50%: 4.054. What does it mean?
Table 6. How is obtained SiO2/(CaO+MgO) ratio?
I am suggesting that CDT data must be normalized to 100 %. Because, through all paper the ratio CDT/PC was used.
Line 105. It is figure 5.
Author Response
Dear Reviewer 1:
Please find attached the answers to your kind and constructive suggestions.
Thank you very much for your input.
Kind regards;
The authors.

Reviewer 3 Report
1. Abstract: contains many unnecessary abbreviations (more than 15 abbreviations), please delete all abbreviations from the abstract (TT, BPF, RD, AD, P, VS, SFST, CATQ, CAP, MCST, UPT, IR, RMS, NDT/PC, etc).
2. Abstract: please re-write the abstract to be more specific. I would prefer to see some data from this study in the abstract, rather than a general description of the results.
3. Abstract: please replace the word “start” by “initial” in the following sentence “start and final setting time” Keywords: Please use semicolons to separate keywords.
4. Introduction: please add and discuss the following references: 1. LV, Zhiqiang; JIANG, Annan; LIANG, Bing. Development of eco-efficiency concrete containing diatomite and iron ore tailings: Mechanical properties and strength prediction using deep learning. Construction and Building Materials, 2022, 327: 126930. 2. YANG, Yingying, et al. Preparation of a novel diatomite-based PCM gypsum board for temperature-humidity control of buildings. Building and Environment, 2022, 226: 109732.
5. Introduction: many unnecessary abbreviations have been used in this manuscript and repeated several times. Therefore, delete all unnecessary abbreviations.
6. Introduction: please correct “(C4AF)” to be “(C4AF)”.
7. Introduction: The novelty of the work versus the literature needs to be explicitly identified.
8. Section 2.2.1: no need to define abbreviation several times in the same subsection “X-ray fluorescence (XRF): X-ray fluorescence (XRF)”
9. Section 2.2.1: Please check "Philips Perl'3 induction".
10. Section 2.2.1 : Please define abbreviation when it introduced for the first time,Wavelength-dispersive X-ray fluorescence (WDXRF)
11. Section 2.2.1: please remove abbrivations from the titles of all sub-sections.
12. Section 2.2.2: please modify the following sentence “The start and final setting time” to be “The initial and final setting time”.
13. Sec. 2.2.3: A new sub-section must be added for specimen preparation, size, curing, ....etc.
14. Sec. 2.2.3: please add chemical and physical composition of the used Portland Cement
15. Sec. 2.2.3: please add specific gravity and water absorption of fine and coarse aggregates
16. Section 3.2 [Fig. 4]: It would be better to compare particle size distribution for diatomite with that for Portland Cement
17. Sec. 3.3: It seems to be a technical report rather than a research article, please discuss the results in detail.

Minor English editing is required. Furthermore, several unnecessary and repeated abbreviations must be deleted.
Author Response
Dear Reviewer 2:
Please find attached the answers to all questions and suggestions made to our work.
Kind regards;
Dr. Jorge L. Costafreda

Round 2
Reviewer 3 Report
The revised manuscript has been read thoroughly and the manuscript has been revised according to the suggestions and comments of the reviewers. The authors’ responses to my comments are satisfactory (they addressed 88% of my comments and failed to adress 12% of my suggestions to improve the revised manuscript). Hence, I recommend that the revised paper may now be accepted for publication in the Materials journal.
Author Response
Dear Reviewer 3:
Thank you very much for your recommendations.
Please find attached the responses to your valuable input.
Kind regards.
Dr. Jorge L. Costafreda.
